# Clinical and cost-effectiveness of DREAMS START (Dementia RElAted Manual for Sleep; STrAtegies for RelaTives) for people living with dementia and their carers: a study protocol for a parallel multicentre randomised controlled trial

Penny Rapaport,[1] Sarah Amador [id] ,[1] Mariam Adeleke,[2] Sube Banerjee,[3] Julie Barber,[2] Georgina Charlesworth,[4,5] Christopher Clarke,[6] Caroline Connell,[6] Colin Espie,[7] Lina Gonzalez,[8] Rossana Horsley,[9] Rachael Hunter [id] ,[8] Simon D Kyle [id] ,[7] Monica Manela,[1] Sarah Morris,[6] Liam Pikett,[1] Malgorzata Raczek,[10] Emma Thornton,[6] Zuzana Walker,[1] Lucy Webster,[1] Gill Livingston[1]

For numbered affiliations see end of article.

**Correspondence to**
Dr Penny Rapaport;
p.rapaport@ucl.ac.uk

## ABSTRACT

**Introduction** Many people living with dementia experience sleep disturbance and there are no known effective treatments. Non-pharmacological treatment options should be the first-line sleep management. For family carers, relatives' sleep disturbance leads to interruption of their sleep, low mood and breakdown of care. Our team developed and delivered DREAMS START (Dementia RElAted Manual for Sleep; STrAtegies for RelaTives), a multimodal non-pharmacological intervention, showing it to be feasible and acceptable. The aim of this randomised controlled trial is to establish whether DREAMS START is clinically cost-effective in reducing sleep disturbances in people living with dementia living at home compared with usual care.

**Methods and analysis** We will recruit 370 participant dyads (people living with dementia and family carers) from memory services, community mental health teams and the Join Dementia Research Website in England. Those meeting inclusion criteria will be randomised (1:1) either to DREAMS START or to usual treatment. DREAMS START is a six-session (1 hour/session), manualised intervention delivered every 1–2 weeks by supervised, non-clinically trained graduates. Outcomes will be collected at baseline, 4 months and 8 months with the primary outcome being the Sleep Disorders Inventory score at 8 months. Secondary outcomes for the person with dementia (all proxy) include quality of life, daytime sleepiness, neuropsychiatric symptoms and cost-effectiveness. Secondary outcomes for the family carer include quality of life, sleep disturbance, mood, burden and service use and caring/work activity. Analyses will be intention-to-treat and we will conduct a process evaluation.

## STRENGTHS AND LIMITATIONS OF THIS STUDY

⇒ This multicentre randomised controlled trial recruiting 370 participants will be the largest, and only fully powered randomised controlled trial of a multicomponent non-pharmacological intervention targeting sleep disturbance in people living at home with dementia to date.

⇒ The study will provide conclusive clinical and cost-effectiveness evidence for the DREAMS START intervention.

⇒ Robust cost-effectiveness data and a mixed-methods process evaluation will, if the intervention is effective, support the potential rollout of the intervention into UK NHS services and other international care settings.

⇒ In our feasibility trial, we found that actigraphy had lower acceptability, and validity than needed as an outcome measure. We are therefore using proxy questionnaires to measure outcomes and using actigraphy as a measure of movement to inform our process evaluation.

⇒ The nature of the intervention is such that participants and those delivering the intervention will not be blinded to treatment allocation.

**Ethics and dissemination** London—Camden & Kings Cross Ethics Committee (20/LO/0894) approved the study. We will disseminate our findings in high-impact peer-reviewed journals and at national and international conferences. This research has the potential to improve sleep and quality of life for people living with dementia and their carers, in a feasible and scalable intervention.

**Trial registration number** ISRCTN13072268.

## INTRODUCTION

The 47 million people living with dementia worldwide is projected to increase to 131 million by 2050; at an estimated trillion US$ cost.[1] Sleep disturbance affects 25%–40% of people living with dementia[2–6] with a meta-analysis finding a pooled prevalence of 26% among people living at home with dementia, 24% of people living with Alzheimer's disease (AD) and 49% of people living with Lewy body dementia.[6] Sleep disturbance affects all aspects of mental and physical functioning and quality of life[7] and may increase the risk of developing or worsening AD.[8] People living with dementia may wake up during the night unaware of the time or be distressed or disorientated. Family members provide most care for the two-thirds of people living with dementia living at home,[9] finding it difficult to cope with persistently disturbed sleep, impacting on quality of life.[10 11] Sleep disturbances also predict family carer depressive symptoms, burden and care home admission, elevating the individual, societal and economic impact of dementia.[10 12 13] Although there are no data currently available for reliable estimates for the costs associated with sleep disturbances and dementia,[14] we do know that sleep disturbances can be a contributing factor to transition to care homes which increases health and social care resource costs. It may also increase the likelihood of falls and loss of weight as people are more tired during the day and this may increase the use of resources.[15]

As with other older people, most people living with dementia have other illnesses and over 90% have at least one long-term health condition and may experience pain, discomfort or mood disturbances[16] which can all contribute to sleep disturbance.[17] Moreover, neurodegeneration of brain structures involved in the regulation of sleep and circadian rhythms, including the suprachiasmatic nucleus, likely mediate changes in sleep timing, sleep continuity and sleep architecture in dementia.[18] Dementia can therefore lead to impaired sleep initiation, reduced night-time sleep, difficulty maintaining sleep, increased night-time wandering and excessive daytime sleepiness.[2 3 19] Treating sleep problems may not only improve well-being, daytime functioning and quality of life of those living with dementia but given the bi-directional relationship between sleep disruption and amyloid deposition and tau pathology[20] could potentially slow disease progression.[21]

There is no conclusive evidence that any therapy to treat sleep disturbance in dementia is effective.[11 14 22] A Cochrane review of pharmacotherapies found no conclusive randomised controlled trial (RCT) evidence for people living with dementia.[23] Antipsychotics and benzodiazepines have adverse effects[24] and increase mortality in older adults.[25] Pharmacological interventions, including melatonin, are not recommended as treatments,[26] and patients and their doctors prefer non-drug approaches for sleep problems.[27] Cochrane reviews have found insufficient evidence for light therapy alone in dementia.[14 28] A recent Cochrane review of non-pharmacological interventions for sleep disturbance in people living with dementia found no conclusive evidence from the 19 included RCTs with none of the included studies identified to be at low risk of bias.[14] This review found some positive effects for carer-focused interventions and for interventions which promoted physical and social activities. Additionally, only four of the included studies recruited people living at home in the community, with most conducted in nursing homes which highlights a gap in good quality evidence conducted with people living with dementia in their own homes. As sleep disturbances in people living with dementia have mixed causes, promising interventions are likely to be multicomponent. Wilfling *et al* conclude that multimodal and complex interventions have the strongest potential to improve sleep disturbance in people living with dementia.[14] Although there is no conclusive evidence of any effective non-pharmacological treatment for sleep disturbances in people living with dementia,[29] two pilot studies in community-dwelling people living with dementia, and our DREAMS START (Dementia RElAted Manual for Sleep; STrAtegies for RelaTives) feasibility RCT, found potential benefits of combining light with other components, including sleep education and hygiene, exercise, daytime activities and cognitive behavioural therapy (CBT).[30–32] We need clinical and cost-effective ways to improve disrupted sleep for people living with dementia, their families, health and social care systems, society and global economies.

This research builds on our successful feasibility RCT (NIHR/HTA 14/220/06).[30 33] Our vision was to synthesise and add to incomplete evidence, co-produce a multicomponent non-pharmacological intervention, and deliver it individually, tailoring it to fit each participant's needs. We co-produced DREAMS START using the best available evidence, patient and public involvement (PPI) and our clinical and research expertise in sleep and dementia.[34] We incorporated the different components, including light, increased activity and exercise and carer's support which show promise in improving outcomes for people living with dementia and sleep disturbance[14] The intervention is delivered to family carers, who implement strategies to reduce the person with dementia's sleep disturbances. It uses natural daylight (where feasible) and if necessary, timed phototherapy to strengthen and stabilise sleep–wake timing. Additionally, it alerts carers to consider pain, uses strategies to increase comfort, reduce anxiety, increase daytime activity and uses CBT for sleep management, which Cochrane reviews found effective for older adults and family carers of people without dementia.[35 36] Our feasibility RCT found that the study design and the intervention were feasible and acceptable; 63/95 (65%; 95% CI 55% to 75%) eligible referrals consented, 62/95 (65%; 95% CI 55% to 75%) were randomised and 37/42 (88%; 95% CI 75% to 96%)

randomised to the intervention adhered to it. Qualitative interviews indicated that it was acceptable.

## Study objectives

This study aims to determine whether our manualised, multicomponent, non-pharmacological treatment package delivered by supervised non-clinically qualified psychology graduates will deliver significant benefits for people living with dementia and their family carers. It is important to demonstrate our intervention is clinical and cost-effective immediately post-intervention and to establish whether this effect is sustained. Since this is an intervention focused on sustaining behaviour change and continuing to use successful strategies, we would anticipate that the effect will be sustained over time and may increase as behaviour change is embedded. This would have a clear impact on individuals and within health services. Therefore, our primary outcome is a measure of sustained effectiveness at 8 months. We have also included analysis at 4 months to establish if the intervention is effective immediately post-intervention. Additional aims are to assess the process and fidelity of delivery of the intervention and explore the experiences of family carers who are the recipients of the intervention sessions to change the sleep of the person with dementia. We will use the data generated to make evidence-based clinical practice recommendations on managing sleep disturbance in dementia.

### Primary objective

To determine whether the DREAMS START intervention improves sleep disturbance in people living with dementia at home at 8 months compared with usual National Health Service (NHS) treatment.

### Secondary objectives

To determine:
1. Whether the DREAMS START intervention improves sleep disturbance in people living with dementia at home at 4 months.
2. Whether it reduces daytime sleepiness (at 4 and 8 months).
3. Whether it improves people living with dementia's quality of life (at 4 and 8 months).
4. The probability that it is cost-effective compared with usual NHS treatment in regard to costs and preference-based health-related quality of life.
5. The role of psychotropic medication and melatonin in any change.
6. Whether it improves the family carer's quality of life (at 4 and 8 months).
7. Whether it improves the family carer's sleep and decreases their affective symptoms and burden (at 4 and 8 months).
8. The mechanisms of change.
9. If effective, how we can optimise the intervention for implementation at scale in the NHS.

## METHODS AND ANALYSIS
### Trial design

This is a multicentre, parallel-group, superiority RCT with masked outcome assessment to test whether the DREAMS START intervention improves sleep disturbance in people living with dementia compared with usual treatment. Participants will be enrolled in the trial for 8 months, assessments will take place at baseline, 4 months, and 8 months post-randomisation (see figure 1 for trial flow diagram). The trial is registered with the ISRCTN and an independent trial steering committee (TSC) and data monitoring and ethics committee provide oversight.

### Participants and recruitment

Potential participant dyads (family carers and people living with dementia) will be recruited from memory services and older adult mental health services in NHS Trusts in London, Essex, Sussex, and Tees, Esk and Wear Valleys and the Join Dementia Research (JDR) website. JDR is a free, secure online and telephone service developed and launched in 2014/2015 by the National Institute of Health and Care Research.

Eligibility will be checked by researchers during initial screening calls to potential participants. People will be included in the study if they meet the following criteria:
1. People living with dementia (any type/severity/on any or no medication), except those currently drinking alcohol heavily.
2. Sleep Disorders Inventory (SDI) score ≥4. The SDI is a valid and reliable standalone tool for sleep disorder in people with dementia.[37] Those who score ≥4 have clinically significant sleep disturbance.
3. Sleep that patient or their family judge as problematic. This is a pragmatic study and if the patient and family are unconcerned, treatment is unnecessary, as is the case in clinical practice. We asked the referrers at the point of entry into the study and assessors asked the family carers and person living with dementia again as part of eligibility criteria by during screening calls and initial assessment meetings.
4. Patient with capacity gives consent, or if not capacitous, consultee gives consent with patient not unwilling.
5. Family carer gives informed consent.
6. Family carer supports the person with dementia emotionally or practically at least weekly.
7. Person with dementia lives in their own home at the beginning of the study with someone present at night.
   Exclusion criteria (any):
1. Known primary sleep breathing disorder diagnosis preceding dementia (eg, sleep apnoea).
2. Current known heavy alcohol use (Alcohol Use Disorders Identification Test—Consumption Score ≥8). We have this criterion as during our earlier feasibility trial two participants drank alcohol during the intervention sessions during the day. These people were unable to work with a plan to change their sleep, which involved reducing alcohol, and we were concerned for the safe-

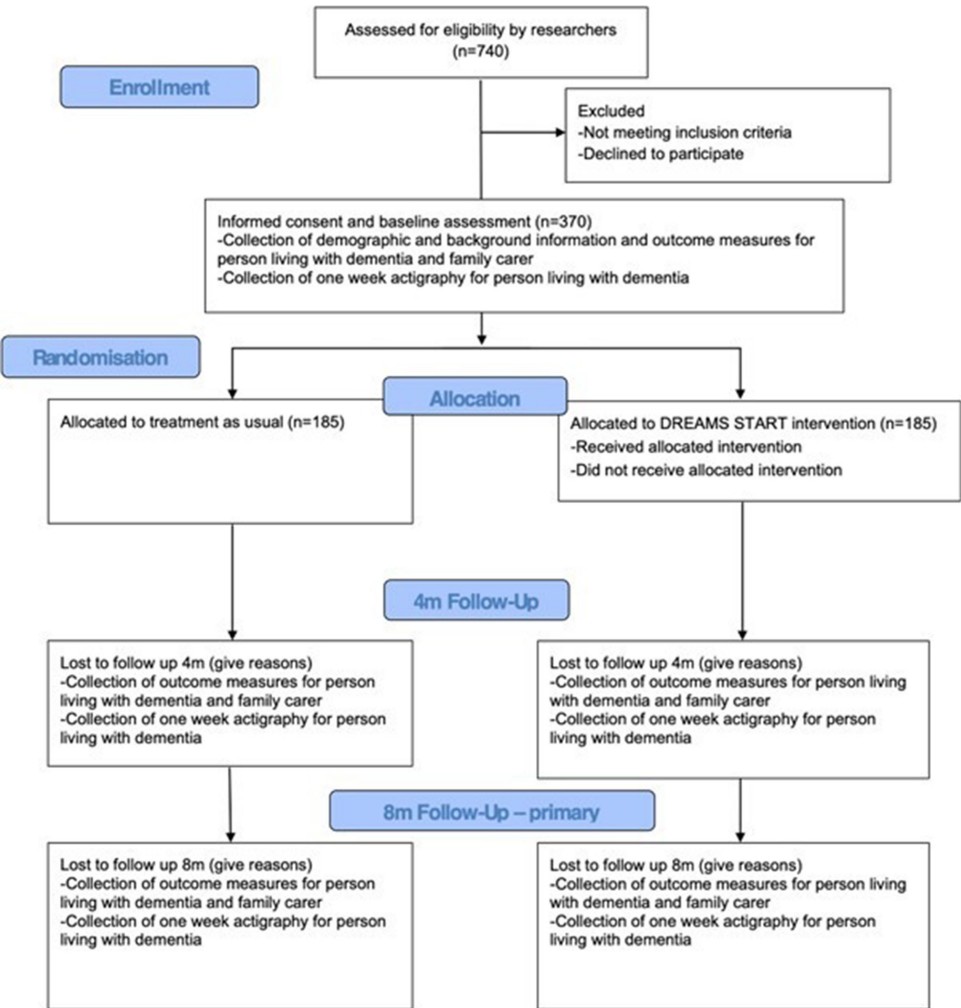

**Figure 1** DREAMS START RCT flow diagram. DREAMS START, Dementia RElAted Manual for Sleep; STrAtegies for RelaTives; RCT, randomised controlled trial.

ty of our facilitators visiting their homes alone. The participants dropped out.

3. People unavailable for >3 weeks (eg, planned holiday or hospital admission) of intervention and follow-up.
4. Currently enrolled in another non-pharmacological dementia RCT.

If potential participants meet eligibility criteria, researchers will offer either a baseline home visit or if necessary, an appointment via telephone or video call. At this appointment, if potential participants are willing, researchers will obtain written informed consent (see online supplemental appendix 1) or where necessary (audio-recorded) verbal consent to take part in the study from family carers and people living with dementia who have the capacity (or a personal consultee will sign for people living with dementia who lack the capacity to consent). They will then complete the baseline assessments (see the section Assessments, and online supplemental table 1).

### Intervention

DREAMS START is a six-session manual-based intervention for carers of people living with dementia to make changes to improve their relatives' sleep.[30 33] People living with dementia can join their carers in sessions if they wish (see online supplemental appendix 2 for a summary of intervention components). The intervention provides information about sleep and dementia, supports carers to use practical *zeitgebers* (cues that influence the person's biological rhythms eg, regular timing of bed and rising, morning wake-up light, regular mealtimes) and to establish adaptive stimulus control (eg, pre-bed settling routine, management of wakeful episodes). It uses strategies to promote de-arousal at night (eg, relaxation, bedroom comfort, no caffeine or alcohol pre-bed, relaxation) and daytime behavioural activation to maintain alertness and reduce daytime naps. The intervention also focuses on helping carers to look after their own (sleep) health, incorporating aspects of CBT. We use actigraphic data from devices worn by participants with dementia at baseline to inform the personalisation of intervention strategies. Over the sessions, carers will build a personalised action plan to continue using after the sessions are completed. We will aim to deliver the sessions weekly or fortnightly depending on the availability of the carer but

will be flexible to maximise adherence. We will deliver either in person, online via video call or by telephone depending on the preferences of the carer.

## Training and supervising facilitators

DREAMS START facilitators will be non-clinically qualified psychology graduates who have participated in a 2-day training programme delivered by the study team online. We have successfully adopted this approach in previous trials.[38–40] This level of expertise ensures the breadth of necessary skills (to impart information, knowledge of dementia, mental health and knowing when to ask for help). Our team has extensive experience in training psychology graduates and psychology assistants in memory and mental health services to deliver non-pharmacological interventions. DREAMS START was designed to be delivered by this same group, to maximise the potential for delivery at scale and at low cost (compared with less available and more expensive qualified psychologists or nurses delivering the intervention) and is compatible with a range of commissioning models. Training will focus on dementia and sleep–wake regulation, empathic listening skills, facilitating behaviour change, interpreting actigraphy, using supervision effectively and working collaboratively with family carers and people living with dementia. Facilitators will receive fortnightly group supervision with a clinical psychologist with additional individual supervision as requested by facilitators or the study team. We used this model in our feasibility RCT and found that the intervention was delivered to a high level of fidelity and was acceptable to and the facilitators positively experienced by participants.

## Usual treatment

We are comparing the new treatment with treatment as usual (TAU). Participants randomised to the new treatment will also receive usual care. TAU varies according to where the person with dementia is treated and their individual needs, but incorporates National Institute for Health and Care Excellence (NICE) guidelines for dementia and consists of assessment, diagnosis, symptomatic interventions, risk assessment and management, advice and information.[26] There is currently no consistent approach to the treatment of sleep difficulties in dementia. We expect usual treatment to reflect that identified during our feasibility trial, where interventions were medical, psychological and social.[30] During our feasibility trial, 45% of participants were prescribed one or more psychotropic medications, with 11% prescribed anxiolytic or hypnotic medication. We will not exclude those taking medication for sleep but will note psychotropic medication prescribed and taken.

## Randomisation, allocation concealment and blinding

The trial manager will perform the randomisation procedure, after consent and baseline data collection are completed. If the trial manager is not available, randomisation will be by a member of the research team who is not involved in participant recruitment or follow-up. Randomisation will be provided by a web-based system using the company Sealed Envelope. Randomisation will occur at the level of the patient and will be blocked and stratified by site using a 1:1 intervention: treatment as usual ratio. Participants will be assigned to treatment groups through consecutive allocation of participant numbers and the use of a Trial Participant Log. The trial manager will notify the intervention facilitators of allocation, who will either arrange the intervention sessions or inform the participant that they have been randomised to the control arm of the trial. Researchers collecting data will be masked to group allocations. The person facilitating the intervention will be different from the researcher conducting the follow-up assessments, if an individual has both roles this will not be in the same research sites. Clinical supervision will be conducted in separate groups based on different research sites to avoid unmasking of intervention facilitators who may also be collecting research data in a different site to that they are delivering in.

Participants and those delivering the intervention will be unmasked and aware of intervention status, which is common in behavioural and psychological intervention trials.[41] As follow-up assessments will be masked, there is a small risk that assessors may become unmasked accidentally by the participant or carer. We will minimise this risk by the following: assessors will remind participants at each stage that they must not discuss their intervention with their assessor and remind participants to hide any study-related materials or equipment. If an assessor does become unmasked, we will make a note of this and ask an alternative assessor to complete future outcome measures for this participant. We have successfully adopted these procedures in previous trials of non-pharmacological interventions.[30 39 42]

## Assessments

The primary outcome is sleep disturbance at 8 months and will be measured using the SDI. The SDI is validated for measuring sleep disturbance in people living with dementia and describes the frequency and severity of sleep-disturbed behaviours. It is validated against clinical variables and used in recent promising pilot studies of non-pharmacological interventions.[30–32 43] The SDI will be collected at baseline, 4 months and 8 months post-randomisation. We originally envisaged the primary outcome in this study would be sleep inferred through actigraphy, which fits with chosen primary outcomes in the recent Cochrane review of non-pharmacological interventions for sleep disturbance in dementia.[14] However, the feasibility study indicated its unsuitability. Actigraphy is not validated in people with dementia and infers sleep from lack of movement. Many carers in our feasibility study informed us it was inaccurate, for example, one participant lay still in bed and screamed at night and actigraph output indicated excellent sleep. Others had movement disorders which are common in some dementias and at times did not lie still although

they and their family thought they were asleep. In addition, markers of going to bed and getting up, which are necessary for actigraphy data interpretation (for at least 7 days; the 'gold' standard), were only available for 68% of people randomised in our earlier study.[33] Conversely, in our feasibility RCT, the SDI was completed at follow-up by 92% of those recruited. This appeared the best way to measure sleep disturbance, and family members in the trial and PPI said that they found it relevant and reflected their difficulties. All secondary outcome measures will be taken at baseline, 4 months and 8 months (see online supplemental table 1).

Secondary outcomes for the person living with dementia are all proxy measures to minimise the burden. These include neuropsychiatric symptoms (Neuropsychiatric Inventory),[44] daytime sleepiness (Epworth sleepiness scale),[45] and quality of life (DEMQOL-Proxy).[46] The DEMQOL-Proxy will also be used in the cost-effectiveness analysis to calculate Quality of life Adjusted Life Years (QALY). A Modified Client Service Receipt Inventory (CSRI)[47] and EQ-5D 5 level (EQ-5D-5L)[48] proxy will be collected to inform the cost-effectiveness analysis and we will also collect details of medication including side effects. We will collect 1 week of actigraphy for the person living with dementia (using the Axivity AX3[49] from baseline and at 4-month and 8-month follow-ups) to inform understandings of the mechanism of change of the intervention. The AX3 is an actigraph, a small, non-invasive device that is worn on the wrist like a watch that measures movement. Typically sleep is inferred from this data from lack of movement. We used actigraphs during our feasibility DREAMS START study and found that they were acceptable to many people living with dementia. We will use the data collected in our process evaluation, exploring changes in activity and how this relates to other outcomes. For family carers, we will assess sleep disturbance (Sleep Condition Indicator),[50] carer anxiety and depression (Hospital Anxiety and Depression Scale),[51] carer burden (Zarit Burden Interview)[52] and health-related quality of life (Health Status Questionnaire).[53] To inform the cost-effectiveness analysis, a Modified CSRI[47] incorporating the Valuation of Informal Care Questionnaire, a measure of carer time and activity and the Brief Work Productivity and Activity Impairment (WPAI), a measure of productivity loss as well as the EQ-5D-5L[48] will be collected at each time point.

## Sample size calculation

We used the SD of baseline SDI scores (15.74) and the correlation between baseline and follow-up measurements (0.57) observed in our feasibility trial for the power calculation.[30] There is no published SDI minimum clinically important difference. We aim to detect a difference of ≥5.5 points, consistent with important differences identified through consultation with experts. This corresponds to a small to medium effect size of 0.35 and is realistic (an average difference of 5.6 was observed in feasibility work). To account for potential facilitator clustering in

the intervention arm, we assumed an intracluster correlation coefficient (ICC) as observed in earlier studies[39 42] (0.03) and approximately 15 participants per facilitator. To achieve equal allocation, the initial calculation, before adjustment for clustering, was based on unequal allocation (1:0.75) which, after inflation for clustering in the intervention arm resulted in 1:1 allocation.

Based on these assumptions, to detect a difference in average SDI of 5.5 between intervention and TAU with power 90% and 5% significance requires 370 participants; 185 in each arm. This calculation allows for up to 15% dropout and includes an inflation for potential non-normality of SDI (as observed in feasibility work).[54] Calculation of sample size was carried out using STATA V.14.

## Adverse events

All AEs (except those listed below as expected) will be recorded in the medical records/case notes/source data in the first instance and will be recorded with clinical symptoms and accompanied by a simple, brief description of the event, including dates as appropriate. All AEs will be recorded until the participant has completed the intervention.

All serious adverse events (SAEs) (except those listed below as expected) will be recorded in the case report form and SAE log. This is because the intervention is not invasive, already has a well-known safety profile and therefore, the collection of AE data will not add any value to the safety profile of the intervention. We will collect information on any planned hospitalisation at baseline and these will not be considered AEs. Based on our feasibility work, we expect few side effects specific to the intervention. The events listed below describe expected procedural/disease-related AEs/SAEs:

► Emotional distress or discomfort as a result of the questionnaires or intervention.
► Falls as a result of increased physical activity or getting out and about more or because of other age-related conditions.
► Death, hospitalisation due to common causes related to dementia (eg, UTIs, delirium, common infections) or transition to a care home as a result of the progression of illness.

## Analysis plan
### Statistical analysis

For each randomised group we will summarise the primary outcome (SDI at 8 months) using means with SD and medians with IQRs. We will also graphically examine the distribution of the score and average scores over time. We will describe the effect of the intervention using the difference in mean SDI score calculated with a 95% CI. This estimate will be obtained from a three-level linear mixed effects regression model which has random effects to allow for repeated outcome measurements at 4 and 8 months and for clustering by facilitator in the intervention arm.[55] The model will include as fixed effects a treatment group indicator, baseline SDI score, study site,

a time indicator and an interaction between treatment group and time. If assumptions of the regression model are violated, the SDI score will be analysed after appropriate transformation (eg, a log transformation). We will calculate the intra-cluster correlation coefficient (with 95% CI) to describe facilitator clustering.

We will carry out all analyses by intention to treat (ITT), comparing the groups as randomised. Analyses of secondary clinical outcomes will take a similar approach to the analysis of the primary outcome. Analyses will be based on people with available outcome data at 4 or 8 months and will rely on an assumption that data is missing at random. We will describe the number (%) of participants with missing outcome in each group, look at reasons for missingness and consider characteristics of the patients excluded from the ITT analysis. In a sensitivity analysis, we will re-estimate the treatment effect with additional adjustments for baseline predictors of missingness. Further analyses based on multiple imputation methods will include the use of pattern mixture models under pre-defined missing not at random scenarios.[56]

A separate statistical analysis plan (available to download from https://doi.org/10.1186/ISRCTN13072268) will predefine the details of all intended analyses.

### Health economic analysis

We will calculate the incremental cost per QALY gained of DREAMS START plus TAU compared with TAU only, over 8 months from a health and social care cost perspective using the EQ-5D-5L proxy to calculate QALYs in line with NICE guidance. Secondary analyses will calculate QALYs using the DEMQOL and relevant tariff and include impact and cost of carers (paid as well as family and close others) out of pocket costs and relevant wider societal costs.

We will include the EQ-5D-5L (a generic measure of health-related quality of life) for carers in the economic evaluation. We will not measure carer productivity loss alone, some carers will be retired, so measuring productivity loss, which focuses on absenteeism from paid employment and presenteeism within paid employment, is not sufficient here. Instead, we will capture the equivalent of 'productivity loss' in our carer population by measuring days off and changes in employment such as moving from full-time to part-time employment and retirement. This will be costed using the human capital approach. The most important component of this analysis will be the impact on carer time and healthcare resource use which will be measured using the Valuation of Informal Care Questionnaire and a modified CSRI respectively. We will also include the brief WPAI questionnaire for caregivers, to ensure we fully capture productivity losses in carers who are in employment.

We will use the number of sessions attended, duration of the session and grade of the staff delivering the session to calculate participant-level costs of DREAMS START, in addition to the training and supervision cost. We modified the CSRI based on data from our feasibility study and will use costs from published sources to cost healthcare resource use.

Carers will complete the questionnaires about resource use, time spent caring and productivity loss in the past 4 months at baseline, 4 months and 8 months. We will calculate QALYs as the area under the curve, adjusting for baseline, over 8 months using the EQ-5D-5L and DEMQOL-Proxy to calculate QALYs. We will report descriptive statistics for resource use, costs and QALYs for baseline, 4 months and 8 months. We will calculate the mean incremental costs and QALYs of DREAMS START compared with usual care over 8 months using patient-level data linear regression, adjusting for site and facilitator clustering in line with the statistical analysis plan, with 95% CIs, cost-effectiveness planes and cost-effectiveness acceptability curves calculated using bootstrapping. We will use the 'net monetary benefit approach' to construct cost-effectiveness acceptability curves to calculate the probability that the intervention is cost-effective for a range of decision thresholds, ranging from £0 to £100 000. We will conduct sensitivity analyses for any assumptions made. We will assess the level and type of missing data and in conjunction with the statistician determine the most appropriate method for accounting for missing data, which is likely to be multiple imputation using chained equations.

The full Health Economics Analysis Plan is available to download from https://doi.org/10.1186/ISRCTN13072268.

### Process evaluation

In line with the updated MRC framework for developing and evaluating complex interventions,[57] we will conduct a process evaluation to explore how the intervention works and in what contexts. This will be informed by our draft logic model (see figure 2) which prespecifies how we expect the intervention to work and what factors may contribute to this.

#### Qualitative analysis

We will purposively sample and conduct qualitative interviews with 15–20 participants (selected for maximum variation, men and women, completers and non-completers, spouses and non-spouse family carers and people living with dementia, across sites) and five staff delivering the intervention. We will ask about their experiences of different elements of the intervention including the content, form and process of delivery, barriers and facilitators to implementing strategies and perceived impact of the intervention for carers and people living with dementia. We will transcribe qualitative data verbatim, anonymise the transcripts and use NVivo V.12. We will use a thematic analytic approach[58] informed by our draft logic model and the Theoretical Domains Framework.[59] The team will organise data into preliminary themes iteratively identifying similarities and differences in the data. We will triangulate qualitative data with fidelity ratings, intervention adherence (session attendance; action plans, strategy

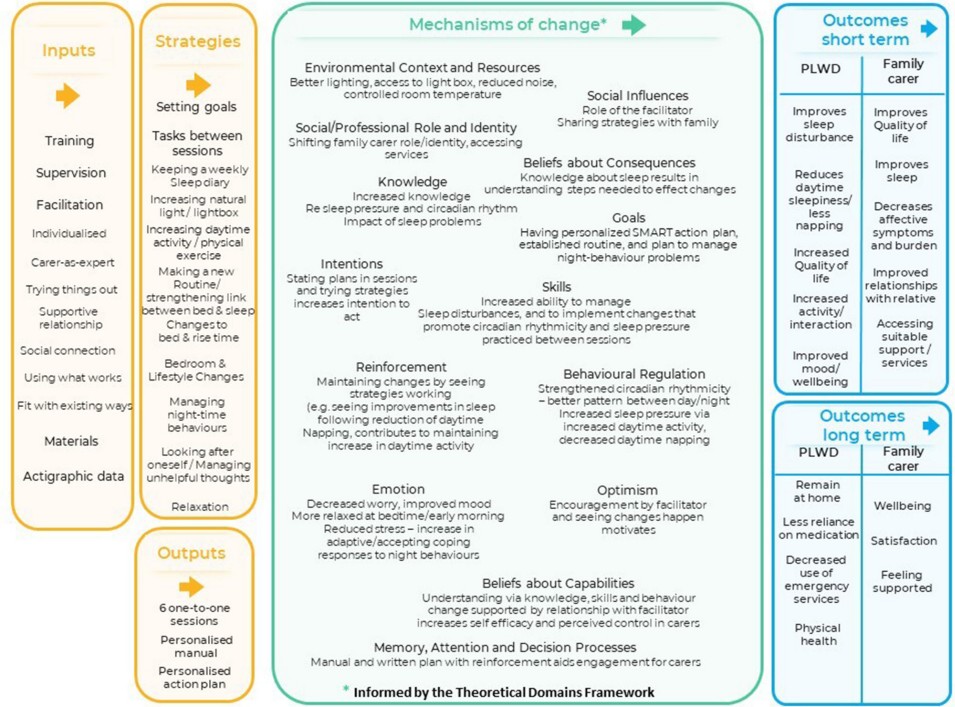

**Figure 2** Draft logic model. PLWD, people living with dementia.

tried) and quantitative outcomes (including patterns in activity from actigraphy).

*Quantitative analysis*

In considering change mechanisms, in line with our draft logic model, the analysis will focus on movement as a potentially important indicator. The intervention incorporates increasing daytime activity and reducing napping to increase pressure to fall asleep at night for the person with dementia as well as strengthening the relationship between bed and sleep at night-time and using light to reinforce circadian rhythmicity. As such, we would expect increased movement during the day and decreased movement at night which we will explore via actigraphy. We will consider rest-activity amplitude, reflecting the relative difference between the least active 5 hours (L5) and the most active 10 hours (M10) in the day. These will be recorded via the actigraphs worn for a week at baseline and for a week at the 4-month and 8-month follow-up points. The mediating effect of these measures on our primary and secondary outcomes will be examined using the stepped approach described by Whittle *et al*.[60] Models fitted in all steps will be mixed models that account for facilitator clustering.

*Fidelity analysis*

To analyse the fidelity of delivery of DREAMS START, we will assess the number of appointments delivered across all intervention participants. Checklists will be applied independently of the facilitator to a random selection of one recorded intervention session for each participant. A mean fidelity score will be produced by dividing the number of items on the checklist identified as being

delivered, by the number of items on the checklist that should have been delivered per appointment, per researcher and across all appointments. We will adopt thresholds used in other intervention fidelity work: where 81%–100% constitutes high fidelity, 51%–80% is moderate fidelity and 50% or lower constitutes low fidelity.[61]

**Patient and public involvement**

We are partnered with the Alzheimer's Society and their research network volunteers, including our co-applicant RH, who has experience as a carer and will continue her input, having contributed to our earlier feasibility study and development of the grant application for this RCT. We have a virtual PPI group and PPI members on our TSC and Trial Management Group, to provide partnership and enhance the relevance, practicality and utility of our materials and findings. They have advised on the language and content of information sheets, contributed to the revision of the intervention having been integral to the initial intervention co-production and are working with us on knowledge exchange and dissemination activities.

**STRENGTHS AND LIMITATIONS**

Few trials have tested interventions for sleep in dementia wholly with those living at home and this will be the largest RCT to date testing a multicomponent, non-pharmacological intervention with those living with dementia and sleep disturbance. There are several potential limitations to our trial protocol. We have excluded

those living alone without someone present at night which reduces the generalisability of our findings to the excluded group, who often have high levels of need and potential risk. This decision was based on our earlier feasibility trial when we included those living alone. When no one else was there at night, carers (family or paid) were unable to implement strategies, like, a scheduled bedtime or wind down routine. It was also difficult to gain reliable information about the sleep patterns of people with dementia living alone and when the intervention was delivered to people living with dementia alone this was often confusing for them. We have also noted above that we could not conduct a double-blind trial of the DREAMS START intervention which potentially increases the risk of performance bias.

## ETHICS AND DISSEMINATION

London—Camden & Kings Cross Research Ethics Committee (Reference: 20/LO/0894) approved the study on 18 September 2020.

We will disseminate our findings in high-impact peer-reviewed journals and at national and international conferences. We will present findings in appropriate local forums for health and social care professionals; participants who have indicated they are interested in the results will be sent a lay summary. We are working with an extensive network of academic researchers, policymakers, stakeholders and PPI, acting as a 'Community of Interest' to advise on and ensure acceptability for the dissemination of the research, and to ensure both national and international reach. We aim to produce an implementation toolkit alongside our DREAMS START intervention and facilitator training materials.

This proposed research has the potential to improve sleep and quality of life for people living with dementia and their family carers, in a feasible and scalable intervention, without medication side effects, and to elucidate the mechanisms of impact. If clinically effective, our intervention should be cost-effective: it is cheap and potentially will reduce the health and social care burden, particularly if it delays care home admission.

## TRIAL STATUS

The trial commenced recruitment in February 2021 and completed recruitment in March 2023 with final primary outcome data collected by November 2023 and the final report will be submitted in May 2024.

**Author affiliations**
[1]Division of Psychiatry, University College London, London, UK
[2]Department of Statistical Science, University College London, London, UK
[3]Faculty of Medicine and Health Sciences, University of Nottingham, Nottingham, UK
[4]Division of Psychology and Language Sciences, University College London, London, UK
[5]North East London NHS Foundation Trust, Rainham, UK
[6]Tees Esk and Wear Valleys NHS Foundation Trust, Darlington, UK
[7]Sleep and Circadian Neuroscience Institute, University of Oxford, Oxford, UK
[8]Research Department of Primary Care and Population Health, University College London, London, UK
[9]Alzheimer's Society, London, UK
[10]Centre for Dementia Studies, Brighton and Sussex Medical School, Brighton, UK

**Contributors** All authors made a substantial contribution to this work. PR drafted the manuscript. All authors were involved in revising the manuscript, giving final approval of the version to be published and agree to be accountable for all aspects of the work. GL and PR conceived of the study and acquired funding. GL, PR, JB, RHu, CCl, SB, MR, GC, ZW, LW, CE, SDK, RHo, SA, MA, LG, MM, SM, ET, CCo and LP contributed to developing the protocol and its implementation.

**Funding** This study/project is funded by the National Institute for Health and Care Research (NIHR) Health Technology Assessment Programme (NIHR HTA 128761). SDK and CAE are supported by the NIHR Oxford Health Biomedical Research Centre.

**Disclaimer** The views expressed are those of the author(s) and not necessarily those of the NIHR or the Department of Health and Social Care.

**Competing interests** The present manuscript is supported by a National Institute for Health and Care Research (NIHR) Health Technology Assessment Programme (HTA) grant (NIHR HTA 128761). SB declares grants from NIHR, CIHR, ESRC, HEE, ESPRC, Alzheimer's Society, and the Alzheimer's Association with no COI with current work. CE declares grants from NIHR-HTA, NIHR-EME, NIHR-BRC, Wellcome Trust with no COI with current work. MR declares a grant from NIHR ARC KSS with no COI with current work. Outside the submitted work SK declares non-financial support from Big Health Ltd. in the form of no cost access to the digital sleep improvement programme, Sleepio, for use in clinical research. Outside the submitted work CE declares stock/stock options and other salary contributions from Big Health Limited developers of Sleepio. Outside the submitted work SSB declares personal fees and non-financial support from Lilly, personal fees from Boehringer-Ingelheim, personal fees from Axovant, personal fees from Lundbeck, personal fees from Nutricia and honoraria from the Hamad Medical Service for lectures and talks. Outside the submitted work MR declares Honorarium for presentation on Lewy body dementias for GE. SB is a trustee of the Alzheimer's Society and NED at the Somerset NHS Foundation Trust. CE is deputy editor of the Journal of Sleep Research and on the editorial board of Sleep Medicine reviews.

**Patient and public involvement** Patients and/or the public were involved in the design, or conduct, or reporting, or dissemination plans of this research. Refer to the Methods section for further details.

**Patient consent for publication** Consent obtained directly from patient(s).

**Provenance and peer review** Not commissioned; externally peer reviewed.

**ORCID iDs**
Sarah Amador http://orcid.org/0000-0003-4196-6410
Rachael Hunter http://orcid.org/0000-0002-7447-8934
Simon D Kyle http://orcid.org/0000-0002-9581-5311

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
