## [Reviewer comments · BMJ Open]

ARTICLE DETAILS

TITLE (PROVISIONAL)	Clinical and cost-effectiveness of DREAMS START (Dementia RElAted Manual for Sleep; STrAtegies for RelaTives) for people living with dementia and their carers. A study protocol for a parallel multi-centre randomised controlled trial.
AUTHORS	Rapaport, Penny; Amador, Sarah; Adeleke, Mariam; Banerjee, Sube; Barber, Julie; Charlesworth, Georgina; Clarke, Christopher; Connell, Caroline; Espie, Colin; Gonzalez, Lina; Horsley, Rossana; Hunter, Rachael; Kyle, Simon; Manela, Monica; Morris, Sarah; Pikett, Liam; Raczek, Malgorzata; Thornton, Emma; Walker, Zuzana; Webster, Lucy; Livingston, Gill

VERSION 1 – REVIEW

REVIEWER	Tang, Yi Xuanwu Hospital, Department of Neurology
REVIEW RETURNED	17-Jul-2023

GENERAL COMMENTS	In this protocol(bmjopen-2023-075273), Rapaport, Penny et al. aimed to develop a randomized controlled trial is to establish whether DREAMS START, a multimodal non-pharmacological intervention is clinically and cost-effective in reducing sleep disturbances in people living with dementia living at home compared to usual care, which will use the data generated to make evidence-based clinical practice recommendations on managing sleep disturbance in dementia. The article is logically coherent and the design is concise. Below are the minor comments. 1. Research aims to intervene in sleep disorders among dementia patients. Therefore, it is necessary to explore the characteristics of sleep disorders in dementia patients and provide a detailed explanation of why this newly proposed intervention method holds potential therapeutic value for sleep disorders in dementia patients. Dementia patients often experience various sleep disturbances, including fragmented sleep, frequent awakenings during the night, and daytime sleepiness. These sleep disturbances can significantly impact their overall well-being and cognitive functioning. The newly proposed intervention method focuses on improving sleep quality and addressing the underlying causes of sleep disorders in dementia patients. It may involve non-pharmacological approaches such as implementing a structured sleep routine, optimizing the sleep environment, promoting relaxation techniques, and providing cognitive-behavioral therapy. These interventions aim to regulate the sleep-wake cycle, reduce sleep fragmentation, and enhance sleep efficiency. The potential therapeutic value of this intervention lies in its ability to improve the quality of life for dementia patients. By addressing sleep
---

	disturbances, it may help alleviate cognitive decline, reduce behavioral symptoms, and enhance overall cognitive functioning. Additionally, better sleep quality can contribute to improved mood, increased alertness during the day, and enhanced caregiver-patient interactions. Further research and clinical trials are needed to validate the effectiveness of this intervention method and determine its long-term benefits for dementia patients with sleep disorders. 2. The first inclusion criterion for this study is " People living with dementia (any type/severity)". One of the outcome measures in this study is " assess the process and fidelity of delivery of the intervention", which requires attention to intervention compliance and difficulty among individuals with severe dementia. This may further impact the effectiveness of the intervention measures, and individuals with severe dementia may struggle to complete them, resulting in the study not proceeding as expected. 3. The potential use of sleeping medications by patients and caregivers may have an impact on research outcomes and influence the exploration of mechanisms, but this point is not found in the inclusion or exclusion criterion. 4. Authors should discuss the basis of "using the Axivity AX3" to inform understandings of the mechanism of change of the intervention carefully.
--	---

REVIEWER	Köpke, Sascha University of Cologne, Institute of Nursing Science
REVIEW RETURNED	23-Jul-2023

GENERAL COMMENTS	The paper presents the protocol for a RCT assessing the effects of the DREAMS START intervention that has been previously developed and assessed in pre-studies. The paper is well written and there are only a few aspects, I recommend addressing for a possible revision. More important aspects: (1) The CONSORT checklist seems not appropriate for a protocol, instead the SPIRIT checklist should be used. (2) I would recommend to present the intervention using the TIDieR template in addition to Appendix 2. (3) Please give information about the status of the trial as it seems to have already started. (4) In the meantime the Cochrane review on "Non-pharmacological interventions for sleep disturbances in people with dementia" (Wilfling et al. 2023) has been published and should be referred to in the background. Importantly procedure and outcomes reported in the review should be compared with those of the planned study. (5) The use of non-clinically qualified facilitators is an important feature of the intervention and the authors should give a rationale for this. This also applies to the relatively short two-day training. (6) From the description on the bottom of p.10, I cannot clearly understand how allocation concealment is guaranteed. Please give more details about the concrete process. (7) Reference to "blinding" and "masking" in this paragraph seems to address the same persons and process. (8) The description of the process evaluation is rather short. Is it based on a framework as e.g. the MRC complex interventions framework (Skivington et al. 2021). Are there specific theoretical foundations? Will a logic model or a theory of change be applied? Especially for the qualitative analysis, I am not sure what will be asked. Could you be more specific here? Further aspects: (1) p.5: "Article Summary": First bullet: What is a "definitive RCT", surely looking at the Cochrane review, it is not the "the only" RCT. In the second bullet: What is "definitive
---

	evidence"? In the third bullet: Why is only implementation in the NHS context mentioned? Are results not applicable to other (international) care settings? (2) p.9.,l.29: Under point 4: Why not also include pharmacological studies? And is only RCT participation meant or participations in clinical studies in general? (3) P.9, l.48ff: This was already stated on p.7. (4) p.10/l.13: I guess the hint to COVID restrictions could be deleted. (5) I do not understand the first sentence on p.11. What or who is separated? (6) On p.12, the bullet points for adverse events are not clear to me, why not refer to "falls" instead of "potential falls" in bullet 2. If I understand bullet 3 correct, these AEs only apply to people living with severe dementia. Why is this? Typos/Errors: (1) p.7/8: Full stops missing at points 8 & 9 of the secondary objectives list. (2) On p.11 under "Assessments" the text is repeated at the end of the first paragraph. (3) p.12/l.40: "ageagerelated".
--	---

REVIEWER	Michalowsky, Bernhard German Center for Neurodegenerative Diseases (DZNE)
REVIEW RETURNED	20-Nov-2023

GENERAL COMMENTS	Thank you for the opportunity to review this study protocol, which addresses an important question. The study is, in general, well-designed. I am attaching my major and minor comments that must be addressed before publishing the protocol (if a publication is allowed since the data assessment is completed this month - 2023 Nov). Minor:  - Abstract: Please state the duration of the intervention. Major:  - The authors state in the background section that sleep disturbance can have an impact on caregiver depression. However, depression is not a secondary outcome. Please state why. - There is also no rationale why improving sleep disturbance can affect the utilization of healthcare resources. Please add a sentence explaining the association between sleep and utilization of HCR. - in the section "study objectives," the authors describe that the primary and secondary aim is to assess the intervention's impact on sleep disturbance after eight months (prior) / four months (secondary). Is this correct? Does this mean all other secondary outcomes are also measured after four instead of 8 months? This is differently explained in the manuscript later on. - Please clearly state what "cost-effective" means in the objective -> do you mean "decrease costs and increase HRQoL"? If yes, please state that in the paper. - Alcohol consumption is an exclusion criterion. No rationale is stated in the background about the association between alcohol consumption and sleep. It would be good to see some background information explaining why alcohol was used as an exclusion criterion. - A very important point is the handling/operationalization of the fourth exclusion criterion: sleep judged as problematic. Please explain how this was handled/operationalized. - There is no limitation section: Please state potential biases, generalizability, and that the inclusion criteria exclude those living alone  representing a biased sample.
---

	 - Is blinding possible? I think PlwD and caregivers of the control group will become aware of their randomization status due to the nature of the intervention when a sleep intervention is missing- Therefore, they will also tell assessors about that fact, which means that blinding is not feasible. - Sample size calculation was based on clusters (n=15 participants per cluster), but only 370 instead of 375 PlwD are needed. It should be 375. - Analysis plan: Please more clearly state how clusters will be handled (random, fixed effects?). I am not a statistician, but I recall that you can not adjust for clusters in repeated measures, like panel data regression, where the time variable is considered. - Why assess resource utilization (retrospectively) at baseline, representing the utilization of healthcare resources before the randomization? This is not needed and is an additional burden to patients and caregivers. - Please add that you will use the "net monetary benefit approach" to create Cost-effectiveness Acceptability Curves. Also, state the WTP thresholds you will use to demonstrate the probability of cost-effectiveness.
--	--

VERSION 1 – AUTHOR RESPONSE

Reviewer 1:

Comments to the Author:

In this protocol(bmjopen-2023-075273), Rapaport, Penny et al. aimed to develop a randomized controlled trial is to establish whether DREAMS START, a multimodal non-pharmacological intervention is clinically and cost-effective in reducing sleep disturbances in people living with dementia living at home compared to usual care, which will use the data generated to make evidence-based clinical practice recommendations on managing sleep disturbance in dementia. The article is logically coherent and the design is concise. Below are the minor comments.

Thank you.

1. Research aims to intervene in sleep disorders among dementia patients. Therefore, it is necessary to explore the characteristics of sleep disorders in dementia patients and provide a detailed explanation of why this newly proposed intervention method holds potential therapeutic value for sleep disorders in dementia patients. Dementia patients often experience various sleep disturbances, including fragmented sleep, frequent awakenings during the night, and daytime sleepiness. These sleep disturbances can significantly impact their overall well-being and cognitive functioning. The newly proposed intervention method focuses on improving sleep quality and addressing the underlying causes of sleep disorders in dementia patients. It may involve non-pharmacological approaches such as implementing a structured sleep routine, optimizing the sleep environment, promoting relaxation techniques, and providing cognitive-behavioral therapy. These interventions aim to regulate the sleep-wake cycle, reduce sleep fragmentation, and enhance sleep efficiency. The potential therapeutic value of this intervention lies in its ability to improve the quality of life for dementia patients. By addressing sleep disturbances, it may help alleviate cognitive decline, reduce behavioral symptoms, and enhance overall cognitive functioning. Additionally, better sleep quality can contribute to improved mood, increased alertness during the day, and enhanced caregiver-patient interactions. Further research and clinical trials are needed to validate the effectiveness of this intervention method and determine its long-term benefits for dementia patients with sleep disorders. We agree with this helpful summary, and we have explained this rationale in our background section.

2. The first inclusion criterion for this study is “ People living with dementia (any type/severity)”. One of the outcome measures in this study is “ assess the process and fidelity of delivery of the intervention”, which requires attention to intervention compliance and difficulty among individuals with severe dementia. This may further impact the effectiveness of the intervention measures, and individuals with severe dementia may struggle to complete them, resulting in the study not proceeding as expected. Thank you for this observation. The intervention is delivered by facilitators to carers of people with dementia and we assess the fidelity of this delivery. We agree that people living with dementia would struggle to implement aspects of the intervention directly and in our earlier feasibility study we found that when we did deliver the intervention to people living with dementia this was the case. Therefore, our inclusion criteria in this RCT stipulated only those who had someone at home with them at night, to support implementation (we have added detail of this under limitations in the revised manuscript). We have also further clarified in our manuscript that the intervention is delivered to carers and that people living with dementia can join sessions: “People living with dementia can join their carers in sessions if they wish.”.

3. The potential use of sleeping medications by patients and caregivers may have an impact on research outcomes and influence the exploration of mechanisms, but this point is not found in the inclusion or exclusion criterion.

As we note in our manuscript: “During our feasibility trial, 45% of participants were prescribed one or more psychotropic medications, with 11% prescribed anxiolytic or hypnotic medication. We will not exclude those taking medication for sleep but will note psychotropic medication prescribed and taken.”. As this is a pragmatic trial, we do not exclude those taking medication, since it is sometimes a part of usual treatment. Instead, we will record this information for both carers and people living with dementia, and we will explore the role of medication (one of our stated secondary objectives) in any observed change within our statistical analysis.

4. Authors should discuss the basis of “using the Axivity AX3” to inform understandings of the mechanism of change of the intervention carefully.

We have added detail on the AX3 and how we will use it under assessments: “The AX3 is an actigraph, a small, non-invasive device that is worn on the wrist like a watch that measures movement. Typically sleep is inferred from this data from lack of movement. We used actigraphs during our feasibility DREAMS START study and found that they were acceptable to many people living with dementia. We will use the data collected in our process evaluation, exploring changes in activity and how this relates to other outcomes.” And in our description of the quantitative process evaluation: “In considering change mechanisms, in line with our draft logic model the analysis will focus on movement as a potentially important indicator. The intervention incorporates increasing daytime activity and reducing napping to increase pressure to fall asleep at night for the person with dementia as well as strengthening the relationship between bed and sleep at night-time and using light to reinforce circadian rhythmicity. As such we would expect increased movement during the day and decreased movement at night which we will explore via actigraphy.”.

Reviewer 2:

The paper presents the protocol for a RCT assessing the effects of the DREAMS START intervention that has been previously developed and assessed in pre-studies. The paper is well written and there are only a few aspects, I recommend addressing for a possible revision.

Thank you.

More important aspects:

(1) The CONSORT checklist seems not appropriate for a protocol, instead the SPIRIT checklist should be used.

We have added a completed SPIRIT checklist and uploaded.

(2) I would recommend to present the intervention using the TIDieR template in addition to Appendix 2.

We have added a completed TIDieR checklist in addition to Appendix 2.

(3) Please give information about the status of the trial as it seems to have already started.

Data collection began in September 2021. We have recently completed our final 8-month (primary time point) data collection to time and target. We plan to publish our main trial findings and the process evaluation in May 2024 and currently our data cleaning processes are underway. We do not know the results. Our intention is for this protocol to be published in advance of the main trial findings. This is detailed under 'Trial Status' in the manuscript. We have checked with the editors of BMJ open that they are still happy in principle to accept this protocol.

(4) In the meantime the Cochrane review on "Non-pharmacological interventions for sleep disturbances in people with dementia" (Wilfling et al. 2023) has been published and should be referred to in the background. Importantly procedure and outcomes reported in the review should be compared with those of the planned study.

Thank you, we have added reference to the Cochrane review to our introduction highlighting that although there are promising areas for intervention, there remains no conclusive evidence for non-pharmacological interventions for sleep disturbance in dementia. We have also added reference to the procedures and outcomes in our methods section to give more rationale for the decisions we made in our trial design and have compared this to the review where appropriate. Although none of the studies in the Cochrane review reported data on quality of life, care home admission, intervention compliance and attrition rates, we are doing so in our study, which we hope will enhance the usefulness of our findings.

We have added to the introduction: "A recent Cochrane review of non-pharmacological interventions for sleep disturbance in people living with dementia found no conclusive evidence from the 19 included RCTs with none of the included studies identified to be at low risk of bias(1). This review found some positive effects for carer focused interventions and for interventions which promoted physical and social activities. Additionally, only four of the included studies recruited people living at home in the community, with most conducted in nursing homes which highlights a gap in good quality evidence conducted with people living with dementia in their own homes."

When describing the intervention development we have added: "We incorporated the different components, including light, increased activity and exercise and carer's support which show promise in improving outcomes for people living with dementia and sleep disturbance(1)".

Within our methods and analysis section, when describing our planned assessments we have added: "We originally envisaged the primary outcome in this study would be sleep inferred through actigraphy, which fits with chosen primary outcomes in the recent Cochrane review of non-pharmacological interventions for sleep disturbance in dementia (1), However, the feasibility study indicated its unsuitability. Actigraphy is not validated in people with dementia and infers sleep from lack of movement. Many carers in our feasibility study informed us it was inaccurate e.g. one participant lay still in bed and screamed at night and actigraph output indicated excellent sleep. Others had movement disorders which are common in some dementias and at times did not lie still although they and their family thought they were asleep. In addition, markers of going to bed and getting up, which are necessary for actigraphy data interpretation (for at least seven days; the "gold" standard), were only available for 68% of people randomised in our earlier study(2). Conversely, in our feasibility RCT, the SDI was completed at follow-up by 92% of those recruited. This appeared the best way to measure sleep disturbance, and family members in the trial and PPI said that they found it relevant and reflected their difficulties."

(5) The use of non-clinically qualified facilitators is an important feature of the intervention and the authors should give a rationale for this. This also applies to the relatively short two-day training. We agree that this is an important feature and have added further rationale for this under training and supervising facilitators: "We have successfully adopted this approach in previous trials(3-5). This level of expertise ensures the breadth of necessary skills (to impart information, knowledge of dementia, mental health and knowing when to ask for help). Our team has extensive experience in training

psychology graduates and psychology assistants in memory and mental health services to deliver non-pharmacological interventions. DREAMS START was designed to be delivered by this same group, to maximise the potential for delivery at scale and at low cost (compared to less available and more expensive qualified psychologists or nurses delivering the intervention) and is compatible with a range of commissioning models” and “We used this model in our feasibility RCT and found that the intervention was delivered to a high level of fidelity and was acceptable to and the facilitators positively experienced by participants.”.

(6) From the description on the bottom of p.10, I cannot clearly understand how allocation concealment is guaranteed. Please give more details about the concrete process.

We have added further detail on this to clarify: “The Trial Manager will perform the randomisation procedure, after consent and baseline data collection are completed. If the Trial Manager is not available, randomisation will be by a member of the research team who is not involved in participant recruitment or follow-up. Randomisation will be provided by a web based system using the company Sealed Envelope. Randomisation will occur at the level of the patient and will be blocked and stratified by site using a 1:1 intervention: treatment as usual ratio. Participants will be assigned to treatment groups through consecutive allocation of participant numbers and the use of a Trial Participant Log. The Trial Manager will notify the intervention facilitators of allocation, who will either arrange the intervention sessions or inform the participant that they have been randomised to the control arm of the trial. Researchers collecting data will be masked to randomisation allocations. The person facilitating the intervention will be different from the researcher conducting the follow-up assessments, if an individual has both roles this will not be in the same research sites. Clinical supervision will be conducted in separate groups based on different research sites to avoid unmasking of intervention facilitators who may also be collecting research data in a different site to that they are delivering in. As follow up assessments will be masked, there is a small risk that assessors may become unmasked accidentally by the participant or carer. We will minimise this risk by the following: assessors will remind participants at each stage that they must not discuss their intervention with their assessor and remind participants to hide any study related materials or equipment. If an assessor does become unmasked we will make a note of this and ask an alternative assessor to complete future outcome measures for this participant.”.

(7) Reference to “blinding” and “masking” in this paragraph seems to address the same persons and process.

We have removed reference to blinding and clarified the process detailed in point 6 above.

(8) The description of the process evaluation is rather short. Is it based on a framework as e.g. the MRC complex interventions framework (Skivington et al. 2021). Are there specific theoretical foundations? Will a logic model or a theory of change be applied? Especially for the qualitative analysis, I am not sure what will be asked. Could you be more specific here?

Yes, we have added further content and included our proposed logic model: “In line with the updated MRC framework for developing and evaluating complex interventions(6) we will conduct a process evaluation to explore how the intervention works and in what contexts, this will be informed by our draft logic model (see Figure 2) which prespecifies how we expect the intervention to work and what factors may contribute to this.” And “We will ask about their experiences of different elements of the intervention including the content, form and process of delivery, barriers and facilitators to implementing strategies and perceived impact of the intervention for carers and people living with dementia.”. We have also highlighted that our thematic analysis will be “informed by our draft logic model and the Theoretical Domains Framework (7).”.

Further aspects:

(1) p.5: “Article Summary”: First bullet: What is a “definitive RCT”, surely looking at the Cochrane review, it is not the “the only” RCT.

We have amended this bullet point which now says: “This multi-centre randomised controlled trial recruiting 370 participants will be the largest, and only fully powered randomised controlled trial of a multicomponent non-pharmacological intervention targeting sleep disturbance in people living at

home with dementia to date". We believe that this is accurate as in the recent Cochrane Review(1), there were only four included studies conducted outside of hospital or long-term care settings. Chan, 2016 tested a Tai Chi intervention with 52 participants, Li, 2009 tested an exercise and activity intervention with 68 participants, McCurry, 2005 tested a multi-component intervention with 36 participants and McCurry, 2011 conducted a multi-component intervention with 132 participants. In the second bullet: What is "definitive evidence"? We have amended this to say 'conclusive' evidence.

In the third bullet: Why is only implementation in the NHS context mentioned? Are results not applicable to other (international) care settings? We agree and have added "and other international care settings" to the point.

(2) p.9,l.29: Under point 4: Why not also include pharmacological studies? And is only RCT participation meant or participations in clinical studies in general? We only include non-pharmacological intervention studies here as we wanted to reduce contamination, for example with overlapping non-pharmacological interventions. We did not include pharmacological trials here as we did not wish to overly restrict our potential recruitment pool. We state here RCT as we were happy for people to participate if they were currently enrolled in an observational study as this would not impact on our trial procedures.

(3) P.9, l.48ff: This was already stated on p.7. We have deleted this repeated point.

(4) p.10/l.13: I guess the hint to COVID restrictions could be deleted. We have deleted this.

(5) I do not understand the first sentence on p.11. What or who is separated? We have added further clarification described above. We are referring to researchers who are both facilitating the intervention in one research site and collecting follow up data at another site, hearing about the allocation of the intervention (for those they are collecting follow up data for) during discussion in clinical supervision.

(6) On p.12, the bullet points for adverse events are not clear to me, why not refer to "falls" instead of "potential falls" in bullet 2. If I understand bullet 3 correct, these AEs only apply to people living with severe dementia. Why is this? This is not the case; we are talking about expected conditions that may occur in our study population. We have clarified the language in the bullet points to make the meaning clearer taking out the words 'possible', 'potential' and removing reference to 'more severe dementia'.

Typos/Errors:

(1) p.7/8: Full stops missing at points 8 & 9 of the secondary objectives list. – Now Added.

(2) On p.11 under "Assessments" the text is repeated at the end of the first paragraph. Now deleted.

(3) p.12/l.40: "ageagerelated". Now corrected.

Reviewer: 3

Thank you for the opportunity to review this study protocol, which addresses an important question. The study is, in general, well-designed. I am attaching my major and minor comments that must be addressed before publishing the protocol (if a publication is allowed since the data assessment is completed this month - 2023 Nov).

Thank you, we have confirmed with the editorial team that publication is still allowed at this time.

Minor:

- Abstract: Please state the duration of the intervention. We have added this information.

Major:

- The authors state in the background section that sleep disturbance can have an impact on caregiver depression. However, depression is not a secondary outcome. Please state why. We are considering caregiver depression – we state in our secondary objectives that we are looking at whether the intervention decreases affective symptoms in caregivers and the Hospital Anxiety and Depression Scale (HADS) is a validated measure of depression and anxiety symptoms we have changed the word 'mood' to 'anxiety and depression' for clarity.

- There is also no rationale why improving sleep disturbance can affect the utilization of healthcare resources. Please add a sentence explaining the association between sleep and utilization of HCR. We have added the following sentence: "Although there is not currently data available for reliable estimates for the costs associated with sleep disturbances and dementia(1), we do know that sleep disturbances can be a contributing factor to transition to care homes which increases health and social care resource costs. It may also increase the likelihood of falls and loss of weight as people are more tired during the day and this may increase use of resources(8).".

- in the section "study objectives," the authors describe that the primary and secondary aim is to assess the intervention's impact on sleep disturbance after eight months (prior) / four months (secondary). Is this correct? Does this mean all other secondary outcomes are also measured after four instead of 8 months? This is differently explained in the manuscript later on. Apologies if this was unclear. We are collecting all secondary outcome measures at four months and eight months and we will be considering the effect of each of the secondary outcomes at each time point (at both 4 and 8 months). We have added to our list of secondary objectives "(at 4 and 8 months)" where relevant for clarification.

- Please clearly state what "cost-effective" means in the objective  do you mean "decrease costs and increase HRQoL"? If yes, please state that in the paper.

To assess the cost-effectiveness, taking into account potential trade-offs between cost and effectiveness, of the intervention when added to usual treatment in improving health-related quality of life for people living with dementia. This includes evaluating whether the intervention is cost-effective compared to usual treatment, acknowledging that cost-effectiveness may involve trade-offs where the intervention is either less expensive and less effective, more expensive and more effective (at or below a decision threshold), or less expensive and more effective (dominant).

Importantly, our study design emphasizes a probabilistic approach, recognizing that the true value of cost-effectiveness parameters is fixed but unknown, and any statistical inference depends on the available evidence. This aligns with the practice of reporting the probability of cost-effectiveness based on available evidence rather than providing deterministic conclusions(9).

We have amended the secondary objective related to cost-effectiveness and it now states: "The probability that it is cost-effective compared to usual NHS treatment in regard to costs and preference based health related quality of life.".

- Alcohol consumption is an exclusion criterion. No rationale is stated in the background about the association between alcohol consumption and sleep. It would be good to see some background information explaining why alcohol was used as an exclusion criterion. Our exclusion criterion is for those people living with dementia currently drinking heavily. We do not exclude those consuming alcohol in moderation which may be impacting on sleep and this is addressed as part of the DREAMS START intervention. The rationale for the exclusion of severe alcohol use came from our feasibility study, when we did not have this exclusion criteria, but for both clinical and safety reasons added the criterion for this RCT. We have added to the text: "We have this criterion as during our earlier feasibility trial two participants drank alcohol during the intervention sessions during the day. These people were unable to work with a plan to change their sleep, which involved reducing alcohol, and we were concerned for the safety of our facilitators visiting their homes alone. The participants dropped out."

- A very important point is the handling/operationalization of the fourth exclusion criterion: sleep judged as problematic. Please explain how this was handled/operationalized. We agree that this is important, and we have added the following text to this criterion: "as is the case in clinical practice. We asked the referrers at the point of entry into the study and assessors asked the family carers and

person living with dementia again as part of eligibility criteria by during screening calls and initial assessment meetings.”

- There is no limitation section: Please state potential biases, generalizability, and that the inclusion criteria exclude those living alone  representing a biased sample.

We have added a brief strengths and limitations section: “Few trials have tested interventions for sleep in dementia wholly with those living at home and this will be the largest RCT to date testing a multi-component, non-pharmacological intervention with those living with dementia and sleep disturbance. There are a number of potential limitations to our trial protocol. We have excluded those living alone without someone present at night which reduces the generalizability of our findings to the excluded group, who often have high levels of need and potential risk. This decision was based upon our earlier feasibility trial when we included those living alone. When no one else was there at night, carers (family or paid) were unable to implement strategies, like, a scheduled bedtime or wind down routine. It was also difficult to gain reliable information about the sleep patterns of people with dementia living alone and when the intervention was delivered to people living with dementia alone this was often confusing for them. We have also noted above that we could not conduct a double-blind trial of the DREAMS START intervention which potentially increases the risk of performance bias.”

- Is blinding possible? I think PlwD and caregivers of the control group will become aware of their randomization status due to the nature of the intervention when a sleep intervention is missing- Therefore, they will also tell assessors about that fact, which means that blinding is not feasible. It is only possible for the trial to be single blind. Within this trial design blinding is not possible for those receiving or delivering the intervention as we tell participants whether they are in the intervention or usual treatment arm of the trial. This is common in psychological and behavioural interventions, where blinding of those delivering and receiving the intervention is challenging both practically and conceptually (10) and we have added further clarification manuscript. Our design does include masked outcome assessment (which we have described further in response to reviewer 2 above) and we believe that the risk of participants telling researchers of their intervention status remains low. We have however taken steps to mitigate against this and have added to our manuscript detail on this: “Participants and those delivering the intervention will be unmasked and aware of intervention status, which is common in behavioural and psychological intervention trials(10). As follow up assessments will be masked, there is a small risk that assessors may become unmasked accidentally by the participant or carer. We will minimise this risk by the following: assessors will remind participants at each stage that they must not discuss their intervention with their assessor and remind participants to hide any study related materials or equipment. If an assessor does become unmasked we will make a note of this and ask an alternative assessor to complete future outcome measures for this participant. We have successfully adopted these procedure in previous trials of non-pharmacological interventions(4, 11, 12).”

- Sample size calculation was based on clusters (n=15 participants per cluster), but only 370 instead of 375 PlwD are needed. It should be 375.

We have updated the text to clarify more detail about how the minimum required sample size was calculated. This incorporated an inflation to allow for approximately 15 participants per cluster in the intervention arm (reduced to 13 before accounting for attrition) while aiming for balanced (1:1) allocation between the groups. Further inflation was applied for attrition (in both arms) and for potential non normality of the SDI score. The approach used was reviewed by the funding body at the outset of the trial.

-Analysis plan: Please more clearly state how clusters will be handled (random, fixed effects?). I am not a statistician, but I recall that you can not adjust for clusters in repeated measures, like panel data regression, where the time variable is considered.

We have now updated this section to reflect the analysis approach subsequently agreed in our separate detailed statistical analysis plan. We have specified the form of the model more clearly adding the text: "This estimate will be obtained from a 3 level linear mixed effects regression model which has random effects to allow for repeated outcome measurements at 4 and 8 months and for clustering by facilitator in the intervention arm (13). The model will include as fixed effects a treatment group indicator, baseline SDI score, study site, a time indicator and an interaction between treatment group and time."

- Why assess resource utilization (retrospectively) at baseline, representing the utilization of healthcare resources before the randomization? This is not needed and is an additional burden to patients and caregivers.

Capturing resource use at baseline is essential for baseline adjustment should there be a random imbalance in high and low resource users between groups, as is best practice(14). It also allows us to provide comprehensive descriptive statistics that capture the baseline scenario, providing a baseline reference for resource utilization patterns. Understanding pre-intervention resource utilization is essential for accurately evaluating the impact of the intervention on healthcare resource consumption. While we acknowledge the potential burden on patients and caregivers, this retrospective assessment is essential for robust research outcomes and informed decision-making.

- Please add that you will use the "net monetary benefit approach" to create Cost-effectiveness Acceptability Curves. Also, state the WTP thresholds you will use to demonstrate the probability of cost-effectiveness.

We will utilize the "net monetary benefit approach" to construct Cost-effectiveness Acceptability Curves. As the purpose of an economic evaluation is to provide information to decision makers not to state if something is cost-effective, choosing a decision threshold is not recommended (and is not something that needs to be reported based on the CHEERS checklist). Instead, we report the probability that the intervention is cost-effective for a range of decision thresholds. We have added the range of decision thresholds to the protocol. There is no willingness to pay threshold available in England as the relevant studies have not been conducted. We have added: "We will utilize the "net monetary benefit approach" to construct cost-effectiveness acceptability curves. On top of this, we will report the probability that the intervention is cost-effective for a range of decision thresholds, ranging from £0 to £100,000."

In summary, we believe that these revisions enhance the paper and hope that you and the reviewers feel we have adequately addressed the suggestions in relation to this research protocol.

Yours sincerely
Dr Penny Rapaport
(On behalf of all authors)

References

1. Wilfling D, Calo S, Dichter MN, Meyer G, Möhler R, Köpke S. Non-pharmacological interventions for sleep disturbances in people with dementia. *Cochrane Database of Systematic Reviews*. 2023(1).
2. Kinnunen KM, Rapaport P, Webster L, Barber J, Kyle SD, Hallam B, et al. A manual-based intervention for carers of people with dementia and sleep disturbances: an acceptability and feasibility RCT. *Health Technology Assessment (Winchester, England)*. 2018;22(71):1-408.
3. Cape J, Leibowitz J, Whittington C, Espie C, Pilling S. Group cognitive behavioural treatment for insomnia in primary care: a randomized controlled trial. *Psychological medicine*. 2016;46(5):1015-25.
4. Livingston G, Barber J, Rapaport P, Knapp M, Griffin M, King D, et al. Clinical effectiveness of a manual based coping strategy programme (START, STRategies for RelaTives) in promoting the

mental health of carers of family members with dementia: pragmatic randomised controlled trial. *BMJ*. 2013;347:f6276.

5. Livingston G, Barber J, Rapaport P, Knapp M, Griffin M, King D, et al. Long-term clinical and cost-effectiveness of psychological intervention for family carers of people with dementia: a single-blind, randomised, controlled trial. *Lancet Psychiatry*. 2014;1(7):539-48.
6. Skivington K, Matthews L, Simpson SA, Craig P, Baird J, Blazeby JM, et al. A new framework for developing and evaluating complex interventions: update of Medical Research Council guidance. *BMJ*. 2021;374:n2061.
7. French SD, Green SE, O'Connor DA, McKenzie JE, Francis JJ, Michie S, et al. Developing theory-informed behaviour change interventions to implement evidence into practice: a systematic approach using the Theoretical Domains Framework. *Implementation Science*. 2012;7.
8. Webster L, Powell K, Costafreda SG, Livingston G. The impact of sleep disturbances on care home residents with dementia: the SIESTA qualitative study. *International Psychogeriatrics*. 2020;32(7):839-47.
9. McFarlane PA, Bayoumi AM. Acceptance and rejection: Cost-effectiveness and the working nephrologist. *Kidney International*. 2004;66(5):1735-41.
10. Juul S, Gluud C, Simonsen S, Frandsen FW, Kirsch I, Jakobsen JC. Blinding in randomised clinical trials of psychological interventions: a retrospective study of published trial reports. *BMJ Evidence-Based Medicine*. 2021;26(3):109-.
11. Livingston G, Barber J, Marston L, Stringer A, Panca M, Hunter R, et al. Clinical and cost-effectiveness of the Managing Agitation and Raising Quality of Life (MARQUE) intervention for agitation in people with dementia in care homes: a single-blind, cluster-randomised controlled trial. *The Lancet Psychiatry*. 2019.
12. Livingston G, Barber JA, Kinnunen KM, Webster L, Kyle SD, Cooper C, et al. DREAMS-START (Dementia RElAted Manual for Sleep; STRategies for RelaTives) for people with dementia and sleep disturbances: a single-blind feasibility and acceptability randomized controlled trial. *International psychogeriatrics*. 2018:1-15.
13. Walwyn R, Roberts C. Therapist variation within randomised trials of psychotherapy: implications for precision, internal and external validity. *Statistical Methods in Medical Research*. 2010;19(3):291-315.
14. Franklin M, Lomas J, Walker S, Young T. An Educational Review About Using Cost Data for the Purpose of Cost-Effectiveness Analysis. *Pharmacoeconomics*. 2019;37(5):631-43.

VERSION 2 – REVIEW

REVIEWER	Tang, Yi Xuanwu Hospital, Department of Neurology
REVIEW RETURNED	03-Jan-2024

GENERAL COMMENTS	In this protocol, the authors introduced DREAMS START, a multi-component non-pharmacological intervention that integrates various elements such as light exposure, increased activity, exercise, and caregiver support. However, I would recommend that the authors consider supplementing the protocol with the following:  1. In the first paragraph of the "Introduction," it is suggested to broaden the scope by encompassing sleep disturbance affect of various types of dementia beyond Alzheimer's disease (AD) and Lewy body dementia (LBD). This would contribute to a more comprehensive discussion. 2. While the adaptability of interventions to suit individual participant needs is commendable, it is essential to assess the impact of this individualization within the framework of the randomized controlled trial (RCT).
---

	3. In the "Study Objectives" section, it is advisable to underscore that the intervention is specifically targeted towards the caregivers of the patients. 4. In examining the secondary objectives, it is recommended to provide clarification on the rationale behind the choice of a four-month duration for assessing the impact of the DREAMS START intervention on sleep disorders in individuals with dementia, as opposed to an eight-month timeframe.
REVIEWER	Köpke, Sascha University of Cologne, Institute of Nursing Science
REVIEW RETURNED	23-Dec-2023
GENERAL COMMENTS	The authors have adequately revised the paper following the reviewers' suggestions.
REVIEWER	Michalowsky, Bernhard German Center for Neurodegenerative Diseases (DZNE)
REVIEW RETURNED	22-Dec-2023
GENERAL COMMENTS	Thanks for considering my comments. Only a minor comment: No one can tell you if an intervention is cost-effective or not (dichotomous) without a defined WTP thresholds. However, you can inform decision makers about the probability of cost-effectiveness at various WTP thresholds (as you said). However, to do so, you have to make use of the bootstrap replications, calculating how many replications would be considered as cost-effective at various WTP thresholds. For this, you always use the NMB approach $[(WTP \times \text{incr. QALY}) - \text{incr. cost} > 0 ?]$, not only to draw the CEACs.

VERSION 2 – AUTHOR RESPONSE

Reviewer: 3

Dr. Bernhard Michalowsky, German Center for Neurodegenerative Diseases (DZNE)

Comments to the Author:

Thanks for considering my comments. Only a minor comment: No one can tell you if an intervention is cost-effective or not (dichotomous) without a defined WTP thresholds. However, you can inform decision makers about the probability of cost-effectiveness at various WTP thresholds (as you said). However, to do so, you have to make use of the bootstrap replications, calculating how many replications would be considered as cost-effective at various WTP thresholds. For this, you always use the NMB approach $[(WTP \times \text{incr. QALY}) - \text{incr. cost} > 0 ?]$, not only to draw the CEACs.

We agree with this reviewer and we will use the net monetary benefit approach to calculate the probability that the intervention is cost-effective for a range of decision thresholds. We have amended the health economics section to make this clearer. It is important to note that England does not have a willingness to pay threshold, only decision thresholds and these can differ by decision maker. Therefore, we have not pre-defined a decision threshold.

The manuscript now states: "We will utilise the "net monetary benefit approach" to construct cost-effectiveness acceptability curves to calculate the probability that the intervention is cost-effective for a range of decision thresholds, ranging from £0 to £100,000."

Reviewer: 2

Prof. Sascha Köpke, University of Cologne

Comments to the Author:

The authors have adequately revised the paper following the reviewers' suggestions.

Thank you

Reviewer: 1

Dr. Yi Tang, Xuanwu Hospital

Comments to the Author:

In this protocol, the authors introduced DREAMS START, a multi-component non-pharmacological intervention that integrates various elements such as light exposure, increased activity, exercise, and caregiver support. However, I would recommend that the authors consider supplementing the protocol with the following:

1. In the first paragraph of the "Introduction," it is suggested to broaden the scope by encompassing sleep disturbance affect of various types of dementia beyond Alzheimer's disease (AD) and Lewy body dementia (LBD). This would contribute to a more comprehensive discussion.

We have changed the introduction to include the pooled prevalence across dementia type:

"Sleep disturbance affects 25-40% of people living with dementia(2-6) with a meta-analysis finding a pooled prevalence of 26% among people living at home with dementia, (24% of people living with Alzheimer's disease (AD) and 49% of people living with Lewy body dementia(6)."

2. While the adaptability of interventions to suit individual participant needs is commendable, it is essential to assess the impact of this individualization within the framework of the randomized controlled trial (RCT).

We agree and are doing so within our qualitative and quantitative process evaluation of the RCT. As outlined in our manuscript, we will be exploring with participants which aspects of the intervention they have used and how, and the impact of this upon sleep disturbances.

3. In the "Study Objectives" section, it is advisable to underscore that the intervention is specifically targeted towards the caregivers of the patients.

We have clarified this in our study objectives section: "Additional aims are to assess the process and fidelity of delivery of the intervention and explore the experiences of family carers who are the recipients of the intervention sessions to change the sleep of the person with dementia."

4. In examining the secondary objectives, it is recommended to provide clarification on the rationale behind the choice of a four-month duration for assessing the impact of the DREAMS START intervention on sleep disorders in individuals with dementia, as opposed to an eight-month timeframe. Since there is currently no definitive RCT evidence for non-pharmacological interventions for sleep difficulties in people living with dementia at home, it is important to demonstrate our intervention is clinically and cost effective immediately post-intervention and to establish whether this effect is sustained. Since this is an intervention focused upon sustaining behaviour change and continuing to use successful strategies, we would anticipate that the effect will be sustained over time and may increase as behaviour change is embedded, this would have clear impact within the health services such as the UK NHS. This was requested by our funders as they were particularly interested in establishing if the intervention has a sustained effect. We have added to our study objectives:

"It is important to demonstrate our intervention is clinically and cost effective immediately post-intervention and to establish whether this effect is sustained. Since this is an intervention focused upon sustaining behaviour change and continuing to use successful strategies, we would anticipate that the effect will be sustained over time and may increase as behaviour change is embedded. This would have clear impact for individual and within health services. Therefore, our primary outcome is a

measure of sustained effectiveness at eight-months We have also included analysis at four-months to establish if the intervention is effective immediately post intervention.”

We hope that you and the reviewers feel we have adequately addressed the suggestions in relation to this research protocol.